# In situ observation of a stepwise [2 + 2] photocycloaddition process using fluorescence spectroscopy

Meng-Fan Wang [1,2], Yun-Hu Deng [1], Yu-Xuan Hong [1], Jia-Hui Gu [1], Yong-Yong Cao[3], Qi Liu [1] ✉, Pierre Braunstein[4] & Jian-Ping Lang [1,2] ✉

Using highly sensitive and selective in situ techniques to investigate the dynamics of intermediates formation is key to better understand reaction mechanisms. However, investigating the early stages of solid-state reactions/transformations is still challenging. Here we introduce in situ fluorescence spectroscopy to observe the evolution of intermediates during a two-step [2 + 2] photocycloaddition process in a coordination polymer platform. The structural changes and kinetics of each step under ultraviolet light irradiation versus time are accompanied by the gradual increase-decrease of intensity and blue-shift of the fluorescence spectra from the crystals. Monitoring the fluorescence behavior using a laser scanning confocal microscope can directly visualize the inhomogeneity of the photocycloaddition reaction in a single crystal. Theoretical calculations allow us to rationalize the fluorescence behavior of these compounds. We provide a convenient strategy for visualizing the solid-state photocycloaddition dynamics using fluorescence spectroscopy and open an avenue for kinetic studies of a variety of fast reactions.

The real-time monitoring of chemical processes using in situ techniques is key to understanding the nature of the intermediates and reaction mechanisms on time scales ranging from years to picoseconds and space scales from meters to angstroms, and it has attracted considerable attention owing to its impact in catalysis, synthesis, energy transformation, sensing, etc.[1–5]. However, the sparsity of direct in situ investigation techniques applied to the initial stages of a reaction/transformation, especially in the solid state, constitute a significant challenge[6,7].

Solid-state [2+2] photocycloaddition reactions of olefins offer a powerful access to cyclic organic molecules with specific configurations that would be difficult to obtain using solution reaction methods[8–15]. Although solid-state [2+2] photocycloaddition reactions are well developed, the limited availability of appropriate analytical techniques, the influence of mixtures of unreacted precursors and

intermediates, etc. make it difficult to in situ monitor the progress of solid-state reactions[16,17]. In particular, when the concentrations of cyclobutane products are very low and ultraviolet (UV) light irradiation maintains for a short time, their accurate monitoring and identification become much more challenge. Consequently, relatively few reports have appeared on the in situ monitoring of specific properties in the course of a photocycloaddition process[18]. However, in situ observation of reactions during the entire transformation is particularly important for understanding the dynamic transitions occurring during photocycloaddition reactions[19].

Analytical techniques applicable to photoreactions include nuclear magnetic resonance spectroscopy (NMR)[20–22], infrared spectroscopy (IR)[13], Raman spectroscopy[23], thermal analysis[24], and X-ray diffraction (XRD)[19,25]. However, only a few of these are suitable in our context. In situ NMR is commonly used to monitor the real-time

[1]College of Chemistry, Chemical Engineering and Materials Science, Soochow University, Suzhou 215123 Jiangsu, People's Republic of China. [2]State Key Laboratory of Organometallic Chemistry, Shanghai Institute of Organic Chemistry, Chinese Academy of Sciences, Shanghai 200032, People's Republic of China. [3]College of Biological, Chemical Science and Engineering, Jiaxing University, Jiaxing 314001 Zhejiang, People's Republic of China. [4]Institut de Chimie (UMR 7177 CNRS), Université de Strasbourg, 4 rue Blaise Pascal - CS 90032, 67081 Strasbourg, France. ✉e-mail: qi.liu@suda.edu.cn; jplang@suda.edu.cn

generation of reaction products in solution but is not easily employed for the identification of solid-state reaction products. IR and Raman spectroscopy are mainly used to characterize the changes in functional groups during the reaction process, but the interconversion of isomers is difficult to identify. Thermal analysis mainly depends on differences in the weight loss of unreacted precursors and different photo-addition products. However, it takes almost 1 h to collect one data point. In addition, excessive heating often triggers the reverse (thermal) reaction of dissociation[24]. The XRD technique includes single-crystal X-ray diffraction (SCXRD) and powder X-ray diffraction (PXRD) and although a number of studies are concerned with single-crystal-to-single-crystal (SCSC) transformations in solid-state [2 + 2] photo-cycloaddition reactions, current reports mainly focus on the initial and final stages of the reaction[26,27], lacking in situ observations. Moreover, when the proportions of cyclobutane products generated by short-time illumination are small, the low spatial and time resolution of the XRD technique makes it impossible to detect the local sites or components[28]. Therefore, more readily accessible in situ methods with high sensitivity and short response times are urgently needed if one aims at understanding the reaction process of photocycloaddition.

Owing to their extremely high sensitivity and good selectivity, fluorescence sensing attracts much attention[29–32]. Generally, pyridyl olefin ligands involved in photochemical [2 + 2] cycloaddition reactions within coordination polymers (CPs) are conjugated and exhibit strong fluorescence[13,33]. Their conjugations and structures are modified in the course of the reaction, which affects their luminescence properties[34,35]. Although a number of studies are concerned with photo-controlled fluorescence in [2 + 2] photocycloaddition reactions, these reports mainly focus on the fluorescence changes at the initial and final stages[35–37], lacking in situ observation of the photoreaction process. Therefore, monitoring the whole solid-state [2 + 2] photo-cycloaddition reaction using in situ fluorescence spectroscopy may

provide deeper insight into structural or composition modifications that occur during the process.

In this work, we introduce, for the first time, a new approach for the in situ observation of the [2 + 2] photocycloaddition process in a one-dimensional (1D) diene-ligand-based CP using fluorescence spectroscopy, and the study of its reaction kinetics (Fig. 1a). This is realized by using our recently reported CP single crystal platform, [Cd$_2$(F-1,3-bpeb)$_2$(3,5-DBB)$_4$] (**CP1**, F-1,3-bpeb = 4,4'-(5-fluoro-1,3-phenylene) bis(ethene-2,1-diyl))dipyridine; 3,5-HDBB = 3,5-dibromobenzoic acid), which exhibits a controllable two-step [2 + 2] photocycloaddition triggered by the combined effect of temperature and irradiation[38]. This CP is transformed into the corresponding dicyclobutane product **CP1-2β** ([Cd$_4$(**2β**)$_2$(3,5-DBB)$_8$], **2β** = syn-3,4,12,13-tetrakis(4-pyridyl)-8,17-bis-fluoro-1,2,9,10-diethano[2.2]metacyclophane) under UV irradiation at 365 nm at 25 °C, while the monocyclobutane compound **CP1-1** ([Cd$_2$(**1**)(3,5-DBB)$_4$], **1** = 4,4'-(3,4-bis(3-fluoro-5-(2-(pyridin-4-yl)vinyl) phenyl)cyclobutane-1,2-diyl)dipyridine) is formed at 365 nm and −50 °C, a temperature which prevents the occurrence of the second step of the [2 + 2] photocycloaddition reaction (Fig. 1b). Because the π-conjugated system is modified during the [2 + 2] photocycloaddition process, all three CPs display fluorescence with different intensities and quantum yields (QYs). Thus, **CP1-1** with a higher intramolecular through-space conjugation (TSC)[39,40] displays the largest intensity fluorescence and QY compared to the other two compounds. The fluorescence spectra are collected for both steps of the [2 + 2] photocycloaddition of **CP1**, and the corresponding kinetics are examined. Thanks to the high sensitivity of fluorescence spectroscopy, even when the yield of the cyclobutane product is very low and **CP1** is irradiated with UV light for a short period of time, the fluorescence emission intensity still changes significantly; but this cannot be observed by NMR or XRD techniques. Furthermore, laser scanning confocal microscopy (LSCM)[41] is used to directly visualize the [2 + 2] photo-cycloaddition in a single crystal, which confirms the non-uniform

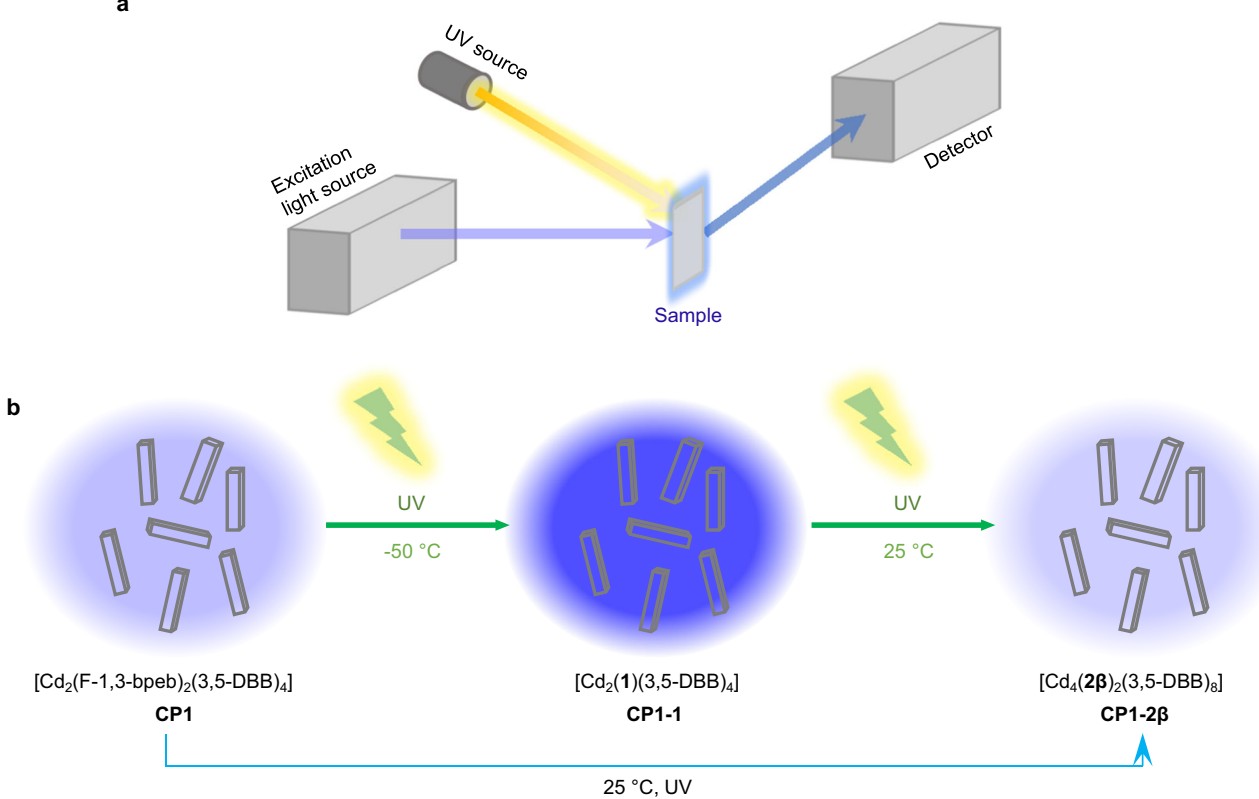

**Fig. 1 | Schematic illustration. a** Schematic diagram of in situ fluorescence spectroscopy. **b** The schematic synthesis routes of the two-step solid-state [2 + 2] photo-cycloaddition reaction.

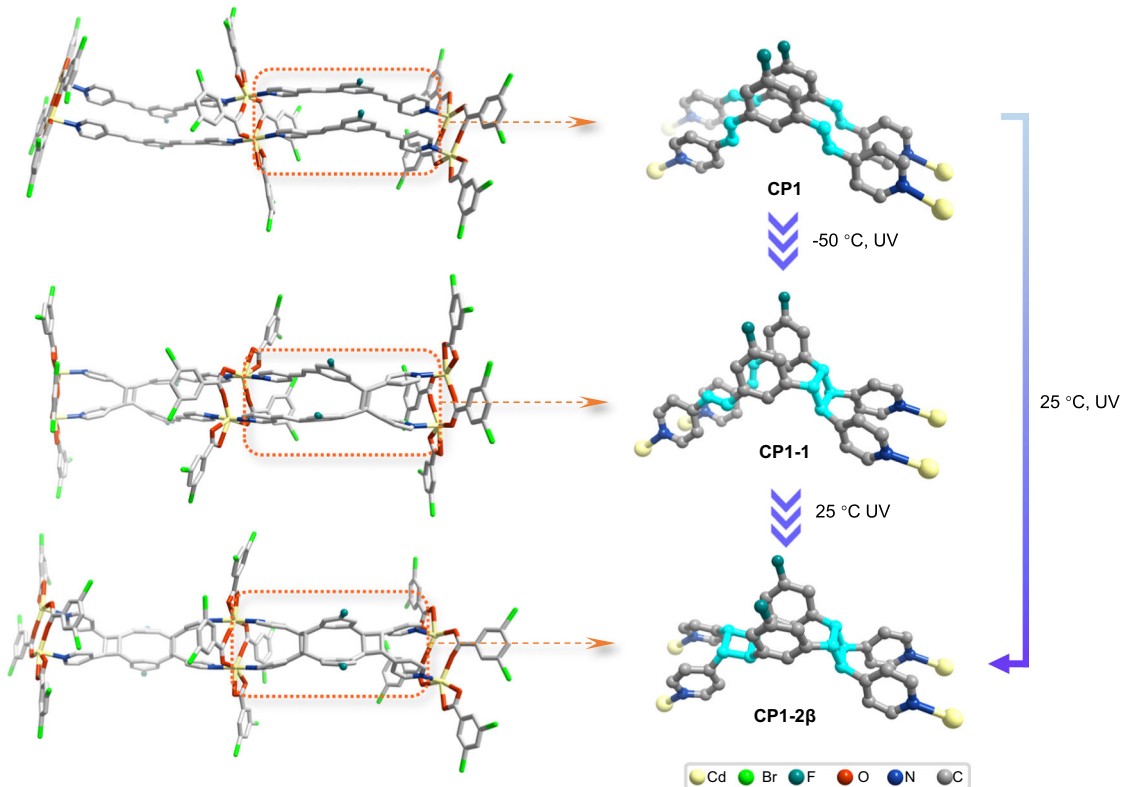

**Fig. 2 | Structures of the two-step [2 + 2] photocycloaddition transformation of CP1 to CP1-2β.** The 1D chain motifs and structures of **CP1**, **CP1-1** and **CP1-2β** illustrate the monomer transformations for each step. For clarity, hydrogen atoms have been omitted. The configurations and transformations of C=C groups associated with the above reactions are highlighted in sky blue.

character of the reaction process between the top and bottom parts of the crystal.

## Results

### Crystal structures and one two-step [2 + 2] photocycloaddition reaction

Colorless crystals of **CP1** were acquired from solvothermal reactions of $3CdSO_4·8H_2O$ with F-1,3-bpeb and 3,5-HDBB according to the previous work[38]. SCXRD analysis revealed that **CP1** crystallizes in the $P\bar{1}$ space group and the asymmetric unit contains a $[Cd_2(F-1.3-bpeb)_2(3,5-DBB)_2]$ unit. **CP1** was previously shown to contain the diene ligands, F-1,3-bpeb, linked by $Cd^{2+}$ ions and second carboxylate ligand 3,5-DBB⁻ to give a one-dimensional (1D) zigzag chain structure. Two pairs of C=C bonds in each set of the adjacent F-1,3-bpeb ligands hold different conformations in the chain of **CP1**. One pair of C=C bonds is aligned to be parallel with a separation of 3.82 Å while the other pair is arranged in a crisscross fashion with a distance of 3.69 Å (Fig. 2 and Supplementary Fig. 1). According to Schmidt's criteria[42], only the pair of parallel C=C bonds in two opposite F-1,3-bpeb ligands can undergo a photocycloaddition reaction.

Our prediction from the C=C arrangement in **CP1** (Fig. 2) is that **CP1-1** would be the photodimerization product from **CP1**. However, **CP1-2β** was obtained after **CP1** got irradiated under UV light (2 W LED lamp, λ = 365 nm) for 1 h at 25 °C while **CP1-1** was generated after **CP1** was irradiated under UV light in 10 min at −50 °C and converted to **CP1-2β** in a further 35 min irradiation at 25 °C in an SCSC fashion, indicating that the C=C groups in the crisscross manner rotates to the parallel position under UV light and 25 °C[43,44]. When **CP1** was exposed to UV light at −50 °C, which greatly blocked the molecular rotation, only the parallel C=C groups experienced dimerization while those arranged in a crisscross manner remained intact, leading to the formation of **CP1-1**. Time-dependent ¹⁹F NMR spectra showed that the monocyclobutane

product was first formed and gradually converted to the dicyclobutane species when **CP1** got irradiated under UV light at 25 °C (Supplementary Fig. 2). SCXRD results, supported by ¹⁹F NMR data, indicated that **CP1-1** can be viewed as an intermediate during the formation of **CP1-2β** from **CP1** (Fig. 2 and Supplementary Figs. 3, 4). The conversion of vinyl ligands to cyclobutane affected photophysical properties of these CPs, which is revealed by the UV-vis adsorption spectra of **CP1** irradiated under UV light at room temperature (Supplementary Fig. 5). Its absorption edge gradually blue-shifted to 390 nm due to the breaking of the π-conjugation of the cyclobutane ligands[45–47].

### In situ fluorescence study

Compounds **CP1**, **CP1-1** and **CP1-2β** are stable in air and retain their crystalline structures intact even when immersed in ethanol for 24 h (Supplementary Fig. 6). These three compounds emitted blue light ($\lambda_{em}$ 451 nm for **CP1**; $\lambda_{em}$ 437 nm for **CP1-1**; $\lambda_{em}$ 437 nm for **CP1-2β**) in the solid state, with QYs of 7.8%, 58.5%, and 1.2% under excitation at 365 nm at room temperature, respectively (Supplementary Fig. 7 and Supplementary Table 1). It showed approximately the same excitation and emission wavelengths as ligand F-1,3-bpeb, suggesting that the emission from **CP1**, **CP1-1**, and **CP1-2β** is of the ligand-to-ligand and intra-ligand charge transfer type (Supplementary Fig. 7).

Collecting the in situ fluorescence spectra of samples of **CP1** continuously irradiated to **CP1-2β** at 25 °C revealed strong variations during the photocycloaddition reaction, the emission intensity increasing by 45 times after irradiation under UV light for 3 min and blue-shifted to 437 nm, followed by a gradual decrease, although it contained both steps of the [2 + 2] photocycloadditions throughout (Fig. 3a, d). It is worth noting that powder samples of **CP1** showed a drastic fluorescence enhancement of 2.7 times after UV light irradiation for only 1 s, while the ¹⁹F NMR data collected after 10 s were still the same as the original ones, indicating the high sensitivity of

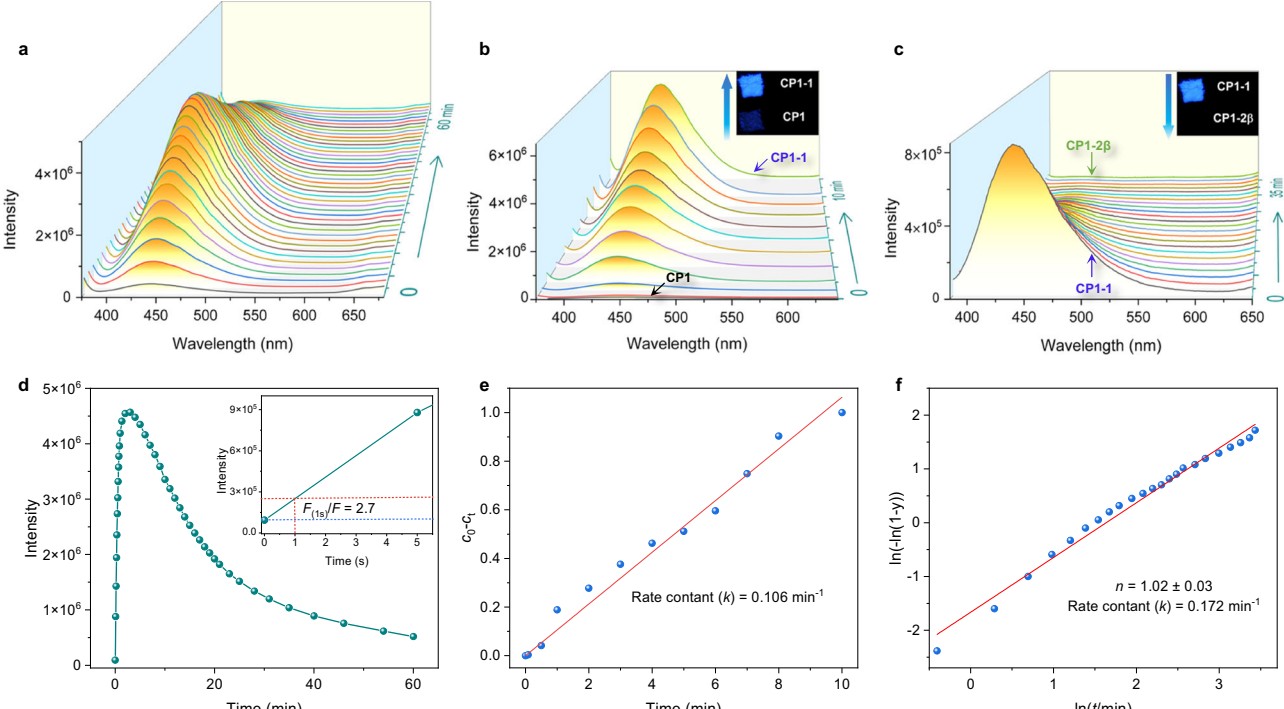

**Fig. 3 | In situ fluorescence spectra. a** The in situ time-dependent fluorescence spectra of **CP1** irradiated with UV light at 25 °C ($\lambda_{ex}$ = 365 nm). **b** The in situ time-dependent fluorescence spectra of **CP1** irradiated with UV light at −50 °C ($\lambda_{ex}$ = 365 nm). Inset: the photographs of **CP1** and **CP1-1** under UV light irradiation. **c** The in situ time-dependent fluorescence spectra of **CP1-1** irradiated with UV light at 25 °C ($\lambda_{ex}$ = 365 nm). Inset: the photographs of **CP1-1** and **CP1-2β** under UV light irradiation. **d** Plots of relative fluorescence emission intensity at 437 nm of **CP1** irradiated under 365 nm at 25 °C ($\lambda_{ex}$ = 365 nm). Inset: Enlargement of the fluorescence emission intensity at 437 nm of **CP1** irradiated at 365 nm and 25 °C. $F$ and $F_{(1s)}$

represent the original fluorescence emission intensity at 437 nm of **CP1** and that after 1 s of 365 nm UV irradiation at 25 °C, respectively. **e** Plots of $c_0$-$c_t$ versus time in the transformation from **CP1** to **CP1-1**. $c_0$ and $c_t$ represent the conversion (mole fraction) calculated from fluorescence intensity data sets of **CP1** before and at any irradiation time ($t$) at 365 nm and −50 °C, respectively. $k$ is the rate constant. **f** Plots of ln(-ln(1-$y$)) versus ln(time) fitted by JMAK model in the transformation from **CP1-1** to **CP1-2β**. $y$ is the conversion (mole fraction) of the photoproduct calculated from fluorescence intensity data sets of **CP1-1** irradiated at 365 nm and 25 °C. $k$ is the rate constant, and $n$ is the dimensionality of growth (Avrami exponent).

fluorescence spectroscopy which allowed changes to be monitored even when the yield of the cyclobutane product was very low in a short time of UV irradiation. The advantage of fluorescence spectroscopy may come from its absolute counting way, which provided intensity as high as order of magnitude of $10^6$ (Fig. 3). Furthermore, a series of fluorescence photographs of this process were recorded as a function of irradiation time. As shown in Supplementary Fig. 8, increases and decreases of fluorescence of the powder of **CP1** were clearly observed, which is consistent with the fluorescence spectral data. Such a phenomenon encouraged us to study systematically the photocycloaddition reaction by in situ fluorescence spectroscopy.

During the [2 + 2] photocycloaddition from **CP1** to **CP1-1** at −50 °C under UV light, the emission intensity increased gradually and blue-shifted to 437 nm in 10 min (Fig. 3b). The emission intensity of **CP1-1** was about 71 times higher than that of **CP1**, and its absolute QY was 58.5%. Subsequently, on going from **CP1-1** to **CP1-2β**, the emission intensity decreased following the UV light irradiation, resulting in about 1/20 of the intensity of **CP1-2β** compared to **CP1-1**, with an absolute QY of 1.2% (Fig. 3c). The fluorescence lifetime ($\tau$) of **CP1, CP1-1** and **CP1-2β** were 6.8 ns, 13.9 ns and 5.4 ns, respectively, confirming the fluorescence feature for all of them (Supplementary Fig. 9).

We researched the kinetics of each step from the results of the in situ time-dependent fluorescence data. The fitting of the conversion data calculated from fluorescence intensity versus irradiation time showed different kinetics for **CP1** to **CP1-1** and **CP1-1** to **CP1-2β**, respectively (Fig. 3e, f). During the transformation from **CP1** to **CP1-1**, a fitting of the conversion percentage of **CP1-1** versus UV light irradiation time at −50 °C resulted in a linear relationship of $c_0$-$c_t$ with the irradiation time, indicating a zero-order behavior[48,49] for the first step from

**CP1** to **CP1-1** with a rate constant of 0.106 min⁻¹, where $c_0$ and $c_t$ represent the conversion (mole fraction) calculated from fluorescence intensity data sets of **CP1** before and at any irradiation time ($t$) at 365 nm and −50 °C, respectively (Fig. 3e). The kinetics of the transformation from **CP1-1** to **CP1-2β** was fitted by applying the Johnson-Mehl-Avrami-Kolmogorov (JMAK) model, which has been successfully applied previously to a number of [2 + 2] photocycloadditions[16,24,50]. The JMAK kinetics are described by Eq. (1):

$$y = 1 - e^{(kt)^n} \tag{1}$$

where $y$ is the conversion (mole fraction) of the photoproduct formed in time $t$, $k$ is the rate constant, and $n$ is the dimensionality of growth (Avrami exponent). The plot of ln(-ln(1-$y$)) versus ln(time) was fitted to attain an Avrami exponent of (1.02 ± 0.03), indicating a first-order behavior for the second step from **CP1-1** to **CP1-2β** with a rate constant of 0.172 min⁻¹ (Fig. 3f). The exponential trends in the mole ratio determined by ¹⁹F NMR closely resemble the trends determined by in situ fluorescence intensity (Supplementary Figs. 10–12). In addition, a linear relationship between the conversion calculated from ¹⁹F NMR and the fluorescence data for both steps could be fitted, indicating that the conversions obtained from in situ fluorescence intensity can be used similarly to those determined by ¹⁹F NMR (Supplementary Fig. 13). The deviation of linearity might be due to the inhomogeneity of [2 + 2] photocycloaddition reaction that originated from non-uniform irradiation geometry, which was observed in the LSCM data.

In order to directly visualize the fluorescence changes within a single crystal during the [2 + 2] photocycloaddition reaction process, LSCM of **CP1** irradiated under UV light was collected at room

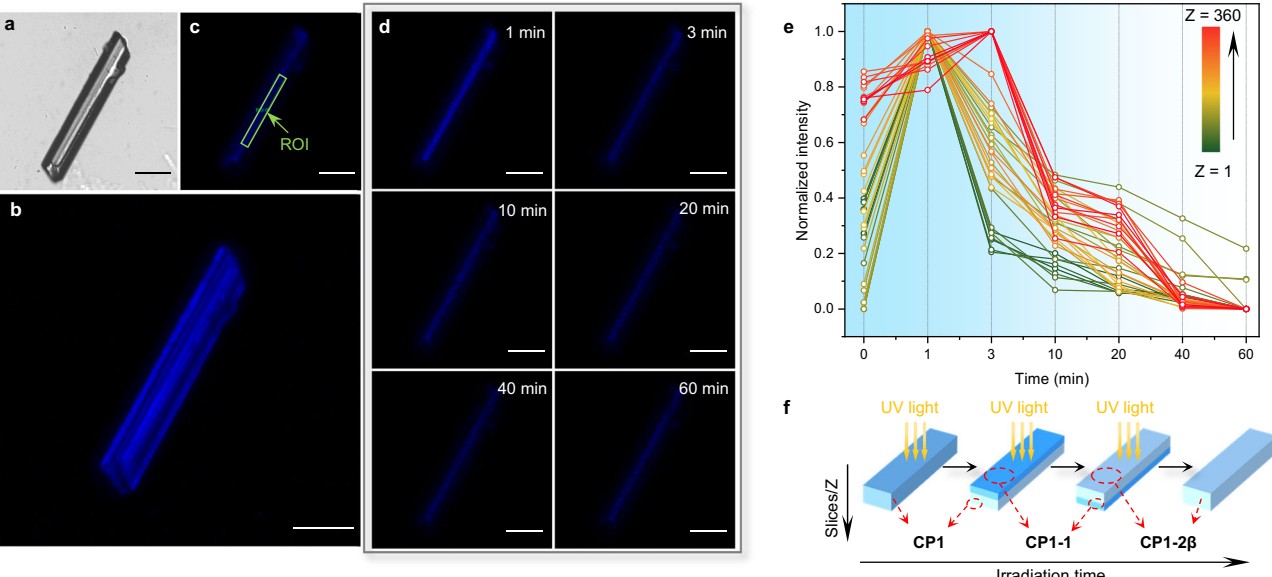

**Fig. 4 | Three-dimensional (3D) LSCM tomograph of CP1. a** Bright-field image and **b** 3D reconstitution of **CP1**, scale bars are 50 μm. **c** LSCM images of **CP1** and **d CP1** irradiated at 25 °C with UV light for some time intervals, at the top first slice, scale bars are 50 μm. **e** Quantified luminescence intensities in (**c**) ROI (regions of interest) range for **CP1** after different irradiation times at different slices. Z represents the different slices in a single crystal. **f** Schematic illustration of the change of fluorescence intensity in a single crystal of **CP1** under UV light irradiation at 25 °C.

temperature and analyzed. The high resolution of LCSM enabled us to observe the details of fluorescence change at each layer of the crystal upon UV light irradiation. Firstly, we examined the reliability of this technique by collecting data for **CP1** without UV light irradiation for 1 h, with every step of 5 s. The blue fluorescence was clearly observed and its intensity was constant (Supplementary Fig. 14).

Then layer scanning from this **CP1** single crystal at different UV light irradiation times ($t = 0$, 1 min, 3 min, 10 min, 20 min, 40 min, and 60 min) at 25 °C was performed from the plane perpendicular to the top of the crystal, with a thickness of 100 nm for each slice (Fig. 4a–d and Supplementary Fig. 15). As shown in Fig. 4d, e and Supplementary Fig. 16, the fluorescence intensity in the same layer first gradually increased and then decreased with irradiation time, consistent with the transformations of **CP1** generating **CP1-1** and **CP1-2β** sequentially, as shown from fluorescence spectra results (Fig. 3). However, the fluorescence intensity was found to change at different depths of the crystal. To gain further insight into the fluorescence intensity changes at different depths of the crystal, we quantified the fluorescence intensities at different depths as a function of exposure time ($t$) by employing the z-stacked scan of LSCM which recorded a series of fluorescence snapshot images (Fig. 4e and Supplementary Fig. 16). As evident in Fig. 4e and f, from the 1st to the 260th slices, i.e., for the top part of the crystal, the fluorescence intensity reached a maximum after 1 min of irradiation, while after the 260th slice (ca. 26 μm deep in the crystal), the highest fluorescence intensity was reached after 3 min of illumination, showing that the photocycloaddition reaction occurred slower in the lower layer than in the top part of the crystal. The photocycloaddition reaction in the single crystal of **CP1** first generated **CP1-1** on the top layers under UV light irradiation at 25 °C, accompanied by increase of fluorescence intensity in this part. As the irradiation went on, the lower part of the crystal began to gradually form **CP1-1**, and the top part was converted from **CP1-1** to **CP1-2β**, accompanied by a brightening in fluorescence of the lower part and a darkening of the top part (Fig. 4f). Monitoring of the [2 + 2] photocycloaddition reaction by following the changes in fluorescence intensity clearly indicated that the [2 + 2] photocycloaddition reaction started from the top of the crystal (UV-exposed side) and gradually reached the bottom layers.

## Mechanism study

The solid-state electron spin resonance (ESR) spectra of **CP1** were measured at room temperature to exclude interference from radical species that have been reported to be responsible for some extrinsic emissions[51]. An electron paramagnetic resonance (EPR) signal was found in **CP1**, and the signal at $g = 2.0024$ is consistent with stable organic radicals (Supplementary Fig. 17). After irradiation under UV light ($\lambda \le 360$ nm) for 2 min and 5 min, no obvious changes were noticed, indicating that radical species were not involved in the fluorescence change of the [2 + 2] photocycloaddition process[52].

Hirshfeld surface analyses of the structures of the repetitive units, [Cd$_4$(F-1.3-bpeb)$_2$(3,5-DBB)$_4$] in **CP1**, [Cd$_4$(**1**)(3,5-DBB)$_4$] in **CP1-1** and [Cd$_4$(**2β**)(3,5-DBB)$_4$] in **CP1-2β**, were conducted and showed similar proportions of intermolecular interactions (Supplementary Figs. 18 and 19). To understand the detailed mechanism of the different QYs associated with **CP1**, **CP1-1** and **CP1-2β**, simplified time-dependent density functional theory (sTDDFT) calculations were utilized to simulate the ground- and excited-state frontier molecular orbitals[39,52]. The molecular orbital surfaces of the highest occupied molecular orbital (HOMO) and lowest unoccupied molecular orbital (LUMO) clearly show that the fluorescence emission band of compound **CP1**, **CP1-1** and **CP1-2β** originates from ligand-to-ligand and intra-ligand charge transfer (Fig. 5). First for **CP1**, in both the ground-state and the excited state, LUMO is mainly on the F-1,3-bpeb ligand, in which there is a large π-conjugation. This means that the fluorescence emission from **CP1** is attributed to the intrinsic through-bond conjugation (TBC)[53] from the F-1,3-bpeb ligands. When **CP1-1** is formed, one pair of C=C bonds reacted to form cyclobutane, partially destroying the electronic conjugation within the F-1,3-bpeb ligands (Supplementary Fig. 18). However, because of the connection between two F-1,3-bpeb ligands, intramolecular TSC was observed between them, with an obvious electronic overlap involving the two adjacent pyridine vinyl groups (LUMO of **CP1-1**) (Fig. 5 and Supplementary Fig. 20). In contrast, two cyclobutanes are formed in **CP1-2β**, which is a nonconjugated organic compound. The nature of the LUMO of **CP1-2β** showed that there is an electronic overlapping between the two adjacent pyridine rings, thereby forming intramolecular TSC in it (Fig. 5 and Supplementary Fig. 20). The larger HOMO-LUMO gaps of **CP1-1**

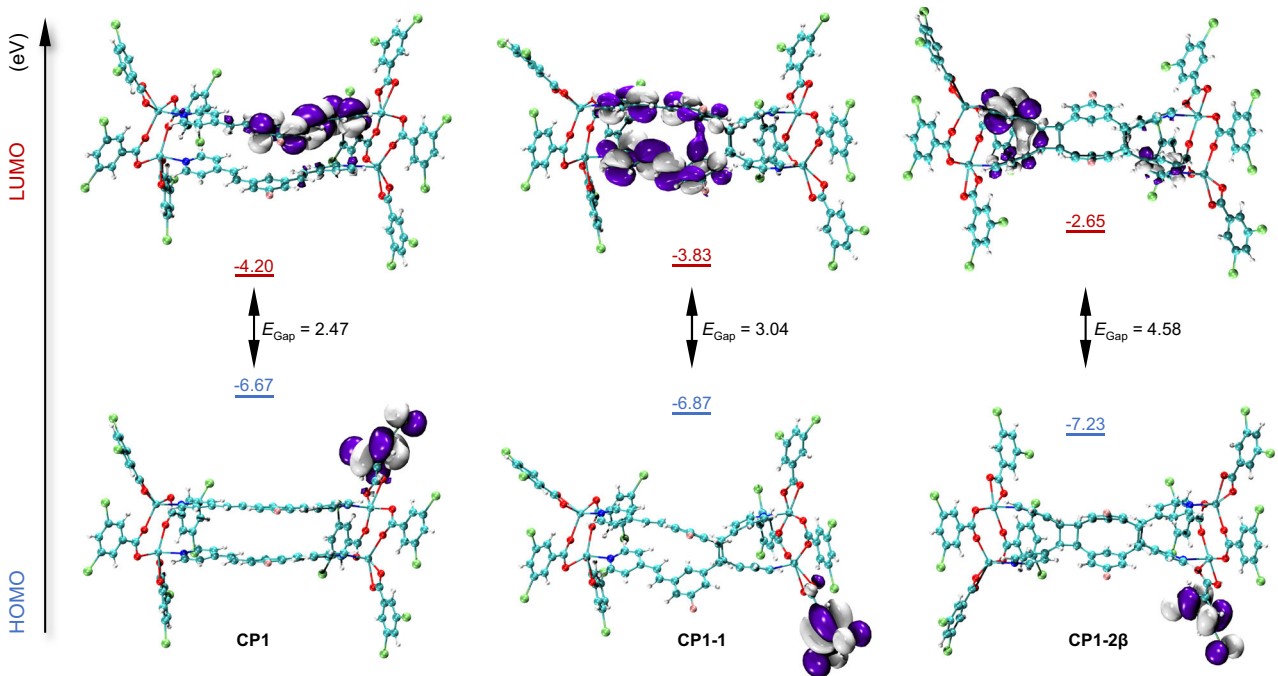

**Fig. 5 | Theoretical calculations of CP1, CP1-1, and CP1-2β.** Frontier molecular orbitals of optimized excited-state geometries of **CP1, CP1-1,** and **CP1-2β** calculated by the sTDDFT method at the PBE0 D3 def2-TZVP level, ORCA 5.0.3 package[63]. $E_{Gap}$ represents calculated energy gap.

and **CP1-2β** compared to **CP1** is also consistent with the blue-shift of the fluorescence spectra on going from **CP1-1** and **CP1-2β** to **CP1**. A combination of intramolecular TSC and TBC in **CP1-1** relative to TBC in **CP1** and intramolecular TSC in **CP1-2β** can account for their different fluorescence properties. In addition, the dimensionless oscillator strength ($f_{osc}$) of **CP1-1** and **CP1-2β** in the crystal state was calculated. The $f_{osc}$ of **CP1-1** and **CP1-2β** from the $S_0$ to $S_1$ states are 0.000054 and 0.0000005, respectively. The larger oscillator strength could help to elucidate the bright fluorescent emission in **CP1-1**, as revealed from fluorescence spectra results and LSCM images (Figs. 3 and 4)[54].

In summary, in situ fluorescence spectroscopy was introduced to monitor the controllable two-step [2 + 2] photocycloaddition process in a CP platform, where the structural changes under UV light irradiation versus time were accompanied by the gradual increase-decrease of intensity and blue-shift of the fluorescence spectra of the CP crystals. This technique showed much higher sensitivity compared to other ones and the related kinetics of each step were investigated, revealing different behaviors for each step, as also confirmed by NMR spectroscopy. In addition, LSCM of a single crystal was performed to directly visualize the change of fluorescence during the [2 + 2] photocycloaddition process. A horizontally uniform and vertically uneven transformation of the crystal was established, depending on the UV light irradiation orientation. Finally, combined theoretical calculations and crystal structure analyses indicated that intramolecular TSC plays an important role in the different fluorescence behaviors of these compounds. This work not only provides a practical strategy for the visualization of [2 + 2] photocycloaddition process but may also open new perspectives for the kinetic study of diverse fast inorganic and/or organic reactions.

## Methods
### Sample preparation for fluorescence measurement
The sample is sandwiched between two quartz pieces for testing. For **CP1** as an example: the original crystals were placed into a mortar and ground into a powder. Next, 3 mg of the powder was dispersed evenly in ethanol (3 mL), and the suspension (100 μL) was dropped on one quartz piece (12 × 20 mm), dried in the air, and then covered with the

other piece of quartz to prepare the sample. The sample thickness was about 0.1 mm.

### Photo-irradiation experiments
The original crystals of **CP1** or **CP1-1** were placed into a mortar and ground into powder. Next, the powder (3 mg) was dispersed evenly in ethanol (3 mL), and the suspension (200 μL) was dropped on one quartz piece (25 × 25 mm), dried in the air to prepare the sample. The sample thickness was about 0.1 mm.

Sample irradiated at room temperature: the sample on the quartz pieces was irradiated with a LED lamp (365 nm, 2 W) for a period of time to form the photoproduct. The distance between the sample and the UV source was fixed to be ca. 10 cm.

Sample irradiated at −50 °C: **CP1** deposited on the quartz pieces was placed in a long glass tube which was immersed in a low-temperature thermostatic reaction bath at −50 °C and irradiated with a LED lamp (365 nm, 2 W) for a period of time to form the photoproduct. The distance between the sample and the UV source was fixed at ca. 17 cm.

### LSCM studies
The block crystals of **CP1** were suspended in ethanol and placed between a pair of glasses. The **CP1** crystals were excited at 405 nm with a semiconductor laser, and the emission was collected at 515–600 nm in the blue channel. The QYs of **CP1, CP1-1,** and **CP1-2β** excited at 405 nm show the same trend as those at 365 nm (QY_{CP1-1} > QY_{CP1} > QY_{CP1-2β}, Supplementary Table 1). The results of two **CP1** single crystals with different sizes were displayed. One was selected for testing focal depth and 3D reconstruction, and the other one was used for time-series scan. The first **CP1** single crystal examined had an irregular hexahedral shape with a size of around 233.4 × 38.1 μm. The z-stack scan was examined from the plane perpendicular to the top of the crystal by 427 slices, with a thickness of 100 nm for each slice, from which a 3D reconstruction of the crystal was generated. All of the images were obtained using the same settings. The external UV light source (365 nm, 2 W) was irradiated from a plane perpendicular to the top of the crystal, and the distance between the crystal and the light

source was ca. 10 cm. The time-series scan of **CP1** single crystal was collected every 5 s until 1 h.

## Computational methods

The theoretical calculations on **CP1**, **CP1-1** and **CP1-2β** were performed for molecules in vacuum. On the basis of the crystal structures, the repetitive units, $[Cd_4(F-1.3-bpeb)_2(3,5-DBB)_4]$ in **CP1**, $[Cd_4(\mathbf{1})(3,5-DBB)_4]$ in **CP1-1** and $[Cd_4(\mathbf{2β})(3,5-DBB)_4]$ in **CP1-2β**, were selected as the initial calculation model. The lowest energy conformations of all compounds at the ground state and the excited state were optimized by B3LYP functional with mixed basis set, 6-311G(d) for C, H, O, N, F, Br, and Lanl2dz for Cd. The XYZ coordinates of optimized geometries are provided as source data in the source file. Gaussian 09 program was used for density functional theory (DFT) calculations[55]. Single-point energy and excited state calculations were implemented by PBE0 hybrid functional with def2-TZVP basis set, def2-J auxiliary basis set, and RI-approximation using ORCA quantum chemistry software (Version 5.0.3)[54,56,57]. Grimme's D3BJ dispersion correction was applied to further describe long-range inter/intramolecular interactions[58]. The relativistic effective core potential (ECP) for Cd was used in all calculations[59]. Vertical excitation energy and orbital distribution of the excited states were calculated through the simplified time-dependent density functional theory (sTDDFT) method at optimized ground state geometry. The distribution of frontier molecular orbitals and electronic transition of singlet states were analyzed on Multiwfn 3.8 software[60]. The Visual Molecular Dynamics (VMD) program was utilized to obtain the color-filled isosurfaces orbitals graphs[61]. The Hirshfeld surfaces and decomposed fingerprint plots were calculated and mapped using CrystalExplorer 21.5 package[62].

## Data availability

Synthetic and experimental procedures, as well as fluorescence spectra, NMR spectra, PXRD, and computational data, are provided in the Supplementary Information. Crystallographic data for the structures reported in this article have been deposited at the Cambridge Crystallographic Data Centre, under deposition numbers CCDC 1889155 (**CP1**), 2036562 (**CP1-1**), and 2036563 (**CP1-2β**). Copies of the data can be obtained free of charge via https://www.ccdc.cam.ac.uk/structures/. Data are also available from the corresponding author upon request. Source data are provided with this paper.

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

## Acknowledgements
The authors thank the National Natural Science Foundation of China (No. 22271203 to J.P.L.), the State Key Laboratory of Organometallic Chemistry of Shanghai Institute of Organic Chemistry (No. KF2021005 to J.P.L.), the Collaborative Innovation Center of Suzhou Nano Science and Technology, the Priority Academic Program Development of Jiangsu Higher Education Institutions, the Project of Scientific and Technologic Infrastructure of Suzhou (No. SZS201905 to J.P.L.), Soochow University Starting Grant (NO. NH10902123 to Q.L.) and Postgraduate Research & Practice Innovation Program of Jiangsu Province (NO. KYCX22_3184 to M.F.W.).

## Author contributions
M.F.W., Q.L. and J.P.L. conceived and designed the experiments. M.F.W. conceived and carried out experiments, determined structures, analyzed data, and corrected the draft of the paper. J.H.G. and Y.X.H. assisted in the fluorescence spectra data. Y.H.D. and Y.Y.C. performed DFT calculations. M.F.W., Q.L., P.B. and J.P.L. analyzed data and wrote the manuscript. All authors contributed to the discussion and revision of the paper.

## Competing interests
The authors declare no competing interests.
