## [Peer Review File · Nature Communications]

In situ observation of a stepwise [2+2] photocycloaddition process using fluorescence spectroscopyReviewers' Comments:

Reviewer #1:

Remarks to the Author:

This manuscript describes the interpretation of fluorescence to gain insight into single-crystal reactivity. The claim is that there are many methods that are not as useful, which is generally true. While the work is interesting, and the system being study is at the high end in terms of complexity, it is confounding that the authors make the claim of usefulness of fluorescence.

There are many examples in which fluorescence is studied with these sorts of reactions - and I note the paper are not cited in the manuscript.

Lead references include:

Li, N.-Y.; Chen, J.-M.; Tang, X.-Y.; Zhanga, G.-P.; Liu, D. Reversible Single-Crystal-to-Single-Crystal Conversion of a Photoreactive Coordination Network for Rewritable Optical Memory Storage. *ChemComm*, 2020, 56, 1984-1987.

Li, N.-Y.; Liu, D.; Ren, Z.-G.; Lollar, C.; Lang, J.-P.; Zhou, H.-C. Controllable Fluorescence Switching of a Coordination Chain Based on the Photoinduced Single-Crystal-to-Single-Crystal Reversible Transformation of a syn-[2.2]Metacyclophane. *Inorg. Chem.* 2018, 57, 849-856.

Papaefstathiou, G.S.; Zhong, Z.; Geng, L.; MacGillivray, L.R. Coordination-Driven Self-Assembly Directs a Single-Crystal-to-Single-Crystal Transformation that Exhibits Photocontrolled Fluorescence. *J. Am. Chem. Soc.* 2004, 126, 9158-9159.

The Zhou reference is particularly important in terms of multiple reactions, while the work of MacGillivray while not a CP still involves monitoring of fluorescence.

The system in the current report is exciting in that the authors study the system in great detail. It would seem that the work may be more suitable for *Angew. Chem.* or *Chem. Sci.* or even *J. Am. Chem. Soc.*

Reviewer #2:

Remarks to the Author:

In this manuscript, the authors report in-situ fluorescence spectroscopy to observe the evolution of intermediates during a two-step solid-state [2+2] photocycloaddition process in a coordination polymer (CP) platform. Each step of the photocycloaddition process could be identified and the kinetics of the photocycloaddition reaction was analyzed. The high sensitivity of fluorescence spectroscopy allows changes to be monitored even when the yield of cyclobutane product is very low and UV light irradiation maintained for a short time, making it advantageous compared to NMR for understanding reaction evolution. Moreover, reaction kinetics and luminous nature of the grown crystals were eventually supported by detailed theoretical investigations as well as laser scanning confocal microscopy. Overall, this work provides a convenient and sensitive strategy for monitoring the product dynamics and reaction mechanism during photoreaction. The results are appealing to broad scientific community which necessitates the surging importance of operando spectroscopy for structure determination. Thus, I recommend its publication after a minor revision.

1. The author should explain the choice of the laser wavelength used in the LSCM studies (405 nm), different than the fluorescence excitation wavelengths (365 nm) of these three compounds.
2. authors are requested to check the stability of each of the CPs in presence of diverse physicochemical conditions. Why have the authors chosen ethanol as a base solvent to crosscheck the stability?

3. In Supplementary Figure 9, the disappearance of characteristic peaks in UV-Vis spectra needs to be justified and experimental conditions should be stated.
4. The statement in mechanistic study section "from the F-1,3-bpeb ligand. When CP1-1 is formed, one pair of C=C bonds reacted to form cyclobutane, partially destroying the electronic conjugation within the F-1,3-bpeb ligands" justify the fluorescence decrement as well quantum yield in CP1-1 than CP1 which is also reflected in the band gap. The authors are requested to check laser scanning confocal microscopic images to support this statement.
5. The X-axis needs to be corrected in Supplementary Figure 5.
6. What do F(1s) and F mean in Figure 2d? Please explain them in the legend.
7. Why NMR data in 10 s did not reflect the reaction for samples of CP1 continuously irradiated to CP1-2 β at 25 °C while a drastic fluorescence enhancement is observed in 1s. This is not well explained.

Reviewer #3:

Remarks to the Author:

The manuscript by Wang and collaborators is related to an alternative analytical methodology to study solid-state reactivity and kinetics, namely fluorescence spectroscopy/microscopy. Specifically, the authors have used fluorescence spectroscopy, in an in situ mode, to study a very well-known solid-state reaction, a [2 + 2] photodimerization, which has been one of the most frequently studied reactions used to investigate fundamental aspects of solid-state reactivity. They also used confocal microscopy to visualize the inhomogeneities caused by the reaction, and the results were supported by computational analysis. Although the authors reported the reaction in reference 34, I am of opinion that this is a fine piece of organic solid-state research that is well executed and carefully presented. The experimental procedures that span several techniques are described in detail. The manuscript is easy to read and conveys the details that could be of interest the solid-state research community because they propose a method that could possibly be used to study other solid-state reactions. I am generally in support of the acceptance of this manuscript. However, my main concern is that the discussion might lack depth in interpretation of the results. After the authors address this and a few other issues outlines below, the manuscript would improve and become acceptable for publication.

Comments:

Page 3, the last paragraph: in addition to the methods enumerated here, the method of photocalorimetry has also been applied to study solid-state cycloadditions (Crystal Growth & Design (2018) 18, 2744), and probably should be mentioned in this context.

Having the chemical formulas in Figure 1 would help the reader to have a quick reference on the chemical structures of the reactant.

Page 6: I was not able to understand whether "mixed" crystals are formed at any of these conditions; the way the text is written, it looks like either one or the other structure of the product is formed. Three distinct reaction conditions are given, however it appears difficult to completely exclude the formation of both in the same crystal. I wonder if there is evidence that excludes the second form? Are there detailed studied of the product composition in respect to the time and temperature? As an example, see JACS (2014) 136, 558, where, due to very complex reactivity that includes various degrees of dimerization, the composition of the reacting mixture was analyzed over time.

Page 6, the first paragraph of the results and discussion section: I am aware that that authors relate this work to a previous work and therefore the structures, except for the two molecules in Figure 1c, are not shown. However, for the readers who are not familiar with the previous work, it might be difficult to visualize and relate the discussion in this section without reference to a figure with the crystal structure that would show these details. I would suggest that the authors include panels with the relevant crystal structure fragments in Figure 1 and refer to this figure when they describe the

criteria for the reactivity.

The discussion on page 7 jumps into the changes of fluorescence in Figure 2, however, we don't see fluorescence spectra of the pure reactants/products. It would be easier to guide the reader through the expected changes in the fluorescence emission if first the spectra of the pure components were shown in Figure 2, and this would also facilitate the discussion on page 7 where the difference between the spectra are discussed.

Figure 2: please include the expressions as well as the model used to calculate the reaction order. There are different models that can be used to model solid-state reaction kinetics that can be used to obtain the order of the reaction.

Figure 2: The plot in panel f shows significant deviation from linearity. Is there an explanation for that? Please comment in the main text.

Page 8: the sentence "while a first-order behavior fitted well for the second step" appears to be grammatically incorrect, and probably should be checked.

Figure 3: This may be a marginal issue, but I am not sure if it is possible to mark the ROI (regions of interest) in Figure 3c (perhaps as an inset for better visualization)?

The molecules studied in this work are very similar to the benzene dicarboxylic acid described in JACS (2009) 131, 7247, where either one or both side chains of benzene dicarboxylic acids can be dimerized, depending on the polymorph or the solvate form. I would suggest that the authors check this reference and perhaps relate their conclusions regarding the mechanisms, which has been studied using different approaches but with a similar goal in the two studies.

I was intrigued by the result presented at the end of page 9 and beginning of page 10. This is indeed the central and probably the most exciting result in this work. However, the discussion here is very short and an in-depth explanation is missing. Can the authors provide deeper discussion that would relate the two clusters of data that can be seen in Figure 3e with the emission spectra of the products? Why there are only two clusters and how does that relate to the relative emission of the two products? What does that tell us of the possible difference in the vertical composition of the product inside the crystal?

The phrase "are subjected to almost the same intermolecular interactions" on page 10 sounds odd and should be rewritten. The compounds can not be subject to interactions; they do include or are composed of entities that interact with each other with intermolecular interactions.

The sentence "Although solid-state photocycloaddition reactions are well developed, the study of their dynamics and reaction mechanisms still represent a challenging field of research" is not correct. There is copious amount of literature that describes both the mechanisms and the dynamics of these reactions.

Minor point: throughout the manuscript, the word "in situ" should be written without the dash.

Point-by-point response to the reviewers' comments on MS ID NCOMMS-23-28696-T

To reviewer #1 (remarks to the author)

This manuscript describes the interpretation of fluorescence to gain insight into single-crystal reactivity. The claim is that there are many methods that are not as useful, which is generally true. While the work is interesting, and the system being study is at the high end in terms of complexity, it is confounding that the authors make the claim of usefulness of fluorescence.

Response: We are grateful to the reviewer for the comment. In the manuscript, we have listed analytical techniques applicable to photoreactions, including NMR, IR, Raman spectroscopy, thermal analysis and XRD, and analyzed their limitations. Therefore, we suggested the necessity to develop more readily accessible in situ methods with high sensitivity and short response times.

This work focuses on in situ fluorescence spectroscopy to identify and analyze the kinetics of each step of the photocycloaddition reaction. The high sensitivity of fluorescence spectroscopy allows changes to be monitored even when the yield of the cyclobutane product is very low and UV light irradiation maintains for a short time. This was evidenced by a drastic fluorescence enhancement of 2.7 times after UV light irradiation for only 1 s. In contrast, no change was detected by NMR or XRD after 10s of reaction (Fig. 3d and Supplementary Fig. 2). Therefore, we believe this work not only provides a practical strategy for the visualization of [2+2] photocycloaddition process but may also play a role for the kinetics study of diverse inorganic and/or organic reactions.

Comment 1: There are many examples in which fluorescence is studied with these sorts of reactions - and I note the paper are not cited in the manuscript. Lead references include:

Li, N.-Y.; Chen, J.-M.; Tang, X.-Y.; Zhang, G.-P.; Liu, D. Reversible Single-Crystal-to-

Single-Crystal Conversion of a Photoreactive Coordination Network for Rewritable Optical Memory Storage. *ChemComm*, 2020, 56, 1984-1987.

Li, N.-Y.; Liu, D.; Ren, Z.-G.; Lollar, C.; Lang, J.-P.; Zhou, H.-C. Controllable Fluorescence Switching of a Coordination Chain Based on the Photoinduced Single-Crystal-to-Single-Crystal Reversible Transformation of a syn-[2.2]Metacyclophane. *Inorg. Chem.* 2018, 57, 849-856.

Papaefstathiou, G. S.; Zhong, Z.; Geng, L.; MacGillivray, L. R. Coordination-Driven Self-Assembly Directs a Single-Crystal-to-Single-Crystal Transformation that Exhibits Photocontrolled Fluorescence. *J. Am. Chem. Soc.* 2004, 126, 9158-9159.

The Zhou reference is particularly important in terms of multiple reactions, while the work of MacGillivray while not a CP still involves monitoring of fluorescence. The system in the current report is exciting in that the authors study the system in great detail. It would seem that the work may be more suitable for *Angew. Chem.* or *Chem. Sci.* or even *J. Am. Chem. Soc.*

Response: We are grateful to the reviewer for the comment. Although there have already been some works about the fluorescence research of [2+2] photocycloaddition reactions, they are mainly focused on fluorescence changes at both initial and final stages. In contrast, we monitored all in situ fluorescence changes during the whole process of two-step [2+2] photocycloaddition reaction. In addition, LSCM was applied to directly observe the fluorescent changes during the [2+2] photocycloaddition reaction of one single crystal. This, for the first time, provided the nonuniformity of the reaction process in the crystal, which is derived from the irradiation geometry.

Thus, we have amended one sentence in the revised MS: “Although a number of studies are concerned with photo-controlled fluorescence in [2+2] photocycloaddition reactions, these reports mainly focus on the fluorescence changes at the initial and final stages,³⁵⁻³⁷ lacking in situ observation of the photoreaction process.” (see Page 4, lines 15-18)

We have also checked the literature carefully and added these references in the References section, while the second reference was cited in the original MS as Ref. 32. (see Pages 19-20, Refs. 35-37 in the revised MS)

To reviewer #2 (remarks to the author)

In this manuscript, the authors report in-situ fluorescence spectroscopy to observe the evolution of intermediates during a two-step solid-state [2+2] photocycloaddition process in a coordination polymer (CP) platform. Each step of the photocycloaddition process could be identified and the kinetics of the photocycloaddition reaction was analyzed. The high sensitivity of fluorescence spectroscopy allows changes to be monitored even when the yield of cyclobutane product is very low and UV light irradiation maintained for a short time, making it advantageous compared to NMR for understanding reaction evolution. Moreover, reaction kinetics and luminous nature of the grown crystals were eventually supported by detailed theoretical investigations as well as laser scanning confocal microscopy. Overall, this work provides a convenient and sensitive strategy for monitoring the product dynamics and reaction mechanism during photoreaction. The results are appealing to broad scientific community which necessitates the surging importance of operando spectroscopy for structure determination. Thus, I recommend its publication after a minor revision.

Response: We highly appreciate the very positive comments of this reviewer on our article.

Comment 1: The author should explain the choice of the laser wavelength used in the LSCM studies (405 nm), different than the fluorescence excitation wavelengths (365 nm) of these three compounds.

Response: We agree with the valuable comment of the reviewer. Although 405 nm equipped in LSCM was not the optimized excitation wavelength of these CPs, we collected the quantum yields of these samples excited at 405 nm, and they showed the same trend as that observed at 365 nm (Page S13, Supplementary Table 1 in revised SI). In this regard, we believe the LSCM data should be consistent to that obtained from in situ fluorescence data. Thus, we have amended the sentence in the revised MS: “The quantum yields of **CP1**, **CP1-1** and **CP1-2 β** excited at 405 nm show the same trend as those at 365 nm ($QY_{CP1-1} > QY_{CP1} > QY_{CP1-2\beta}$, Supplementary Table 1).” (see Page 15, bottom to lines 10-11)

Comment 2: authors are requested to check the stability of each of the CPs in presence of diverse physicochemical conditions. Why have the authors chosen ethanol as a base solvent to crosscheck the stability?

Response: We agree with the valuable comment of the reviewer. PXRD patterns of all three CPs have been collected, as shown in Supplementary Fig. 6. And the experimental data matched well with those simulated from their corresponding single crystal data. It is worth noting that **CP1-1** and **CP1-2 β** were derived from **CP1** and **CP1-1** under UV irradiation, respectively, confirming the photo stability of these CPs. In order to promote the processability of the samples and ensure the UV irradiation at the same conditions, ethanol was applied to disperse CP powders during the fabrication. Thus, the PXRD data of CPs in ethanol were collected, which also showed the same patterns as those simulated ones.

Comment 3: In Supplementary Figure 9, the disappearance of characteristic peaks in UV-vis spectra needs to be justified and experimental conditions should be stated.

Response: We agree with the valuable comment of the reviewer. Upon UV light irradiation, **CP1** got gradually converted into **CP1-1** and **CP1-2 β** , accompanied by the breaking of the π -conjugation of F-1,3-bpeb in the structure. This led to gradual blue-shift of absorption edge in the UV-vis adsorption spectra of **CP1**. Similar observation was also reported in other [2+2] photo-cycloaddition systems (*J. Am. Chem. Soc.* **142**, 8862-8870 (2020), *Angew. Chem. Int. Ed.* **58**, 14865-14870 (2019), *Angew. Chem. Int. Ed.* **58**, 2423-2427 (2018)).

According to the suggestion, the explanation of UV-vis adsorption spectra of **CP1** irradiated under UV light ($\lambda = 365$ nm) at room temperature has been provided in the revised MS: "The conversion of vinyl ligands to cyclobutane affected photophysical properties of these CPs, which is revealed by the UV-vis adsorption spectra of **CP1** irradiated under UV light at room temperature (Supplementary Fig. 5). Its absorption edge gradually blue-shifted to 390 nm due to the breaking of the π -conjugation of the cyclobutane ligands.⁴⁵⁻⁴⁷" (see Page 7, lines 19-23)

And the experimental conditions have been described in the revised SI. (see Page S4,

lines 19-21)

Comment 4: The statement in mechanistic study section “from the F-1,3-bpeb ligand. When CP1-1 is formed, one pair of C=C bonds reacted to form cyclobutane, partially destroying the electronic conjugation within the F-1,3-bpeb ligands” justify the fluorescence decrement as well quantum yield in CP1-1 than CP1 which is also reflected in the band gap. The authors are requested to check laser scanning confocal microscopic images to support this statement.

Response: We are grateful to the reviewer for the constructive suggestion. Although the electronic conjugation within F-1,3-bpeb ligands is partially destroyed when CP1-1 is formed from CP1, an intramolecular through-space conjugation (TSC) is formed between the two adjacent F-1,3-bpeb ligands. This leads to increment of fluorescence intensity and quantum yield of CP1-1 compared to CP1. The larger band gap of CP1-1 versus CP1 is reflected by its blue-shift in emission peak (437 nm vs 451 nm for CP1-1 and CP1, respectively). And this is also consistent with laser scanning confocal microscopic data, where the fluorescence intensity increased firstly when CP1 got transformed to CP1-1 (Fig. 4). (see Page 11, lines 11-15 in the revised MS)

Comment 5: The X-axis needs to be corrected in Supplementary Figure 5.

Response: We thank the reviewer for the careful reviewing. In this work, chemical shifts of ^{19}F NMR were recorded in parts per million (ppm, δ) relative to CFCl_3 ($\delta = 0.00$). The electronic density of F atoms in CP1, CP1-1 and CP1-2 β are greater than that of CFCl_3 , which means their signals will appear in the high field. Therefore, the chemical shifts of the ^{19}F NMR spectra are all negative. The X-axes in Supplementary Figures 2-4 have been modified in the revised SI. (see Pages S7-S8)

Comment 6: What do F(1s) and F mean in Figure 2d? Please explain them in the legend.

Response: We thank the reviewer for the constructive reviewing. F and $F_{(1s)}$ represent the original fluorescence emission intensity of CP1 at 437 nm and that obtained after 1 s of 365 nm UV irradiation at 25 °C, respectively. The above information has been

amended to the legend of Fig. 3d in the revised MS. (see Page 10, lines 6-11)

Comment 7: Why NMR data in 10 s did not reflect the reaction for samples of CP1 continuously irradiated to CP1-2 β at 25 °C while a drastic fluorescence enhancement is observed in 1s. This is not well explained.

Response: We are thankful for the reviewer's valuable suggestion. In our work, when CP1 was exposed to UV light, it underwent a rapid photocycloaddition reaction and was transformed to CP1-1, accompanied by an obvious enhancement of fluorescence intensity in only 1 s. In contrast, NMR data in 10 s still did not reflect this conversion, indicating the high sensitivity of the fluorescence spectroscopy. This difference may come from their different counting methods. Fluorescence spectroscopy collects the absolute counts of emission light, where its intensity is as high as order of magnitude of 10^6 . In contrast, NMR data are relative counts (calculated from the relative ratios of the integrations of the corresponding peaks) and just have order of magnitude of ca. 10^2 (with maximum of 100%). Thus, fluorescence spectroscopy has a much better resolution compared to NMR.

We have modified the following sentences in the revised MS: "It is worth noting that the powder samples of CP1 showed a drastic fluorescence enhancement of 2.7 times after UV light irradiation for only 1 s, while the ^{19}F NMR data collected after 10 s were still the same as the original ones, indicating the high sensitivity of fluorescence spectroscopy, which allowed changes to be monitored even when the yield of the cyclobutane product was very low in a short time of UV irradiation. The advantage of fluorescence spectroscopy may come from its absolute counting way, which provided intensity as high as order of magnitude of 10^6 (Fig. 3)." (see Page 8, lines 7-14)

To reviewer #3 (remarks to the author)

The manuscript by Wang and collaborators is related to an alternative analytical methodology to study solid-state reactivity and kinetics, namely fluorescence spectroscopy/microscopy. Specifically, the authors have used fluorescence spectroscopy, in an in situ mode, to study a very well-known solid-state reaction, a [2

+ 2] photodimerization, which has been one of the most frequently studied reactions used to investigate fundamental aspects of solid-state reactivity. They also used confocal microscopy to visualize the inhomogeneities caused by the reaction, and the results were supported by computational analysis. Although the authors reported the reaction in reference 34, I am of opinion that this is a fine piece of organic solid-state research that is well executed and carefully presented. The experimental procedures that span several techniques are described in detail. The manuscript is easy to read and conveys the details that could be of interest the solid-state research community because they propose a method that could possibly be used to study other solid-state reactions. I am generally in support of the acceptance of this manuscript. However, my main concern is that the discussion might lack depth in interpretation of the results. After the authors address this and a few other issues outlines below, the manuscript would improve and become acceptable for publication.

Response: We highly appreciate the very positive comments of this reviewer on our article and address all the concerns of the reviewer as below.

Comment 1: Page 3, the last paragraph: in addition to the methods enumerated here, the method of photocalorimetry has also been applied to study solid-state cycloadditions (Crystal Growth & Design (2018) 18, 2744), and probably should be mentioned in this context.

Response: We are grateful to the reviewer for the constructive suggestion. We have amended the photocalorimetry method and explained it in the Introduction section in the revised MS: “Thermal analysis mainly depends on differences in the weight loss of unreacted precursors and different photo-addition products. However, it took almost 1 h to collect one data point. In addition, excessive heating often triggers the reverse (thermal) reaction of dissociation.²⁴” (see Page 3, bottom to lines 1-3; Page 4, line 1)

Comment 2: Having the chemical formulas in Figure 1 would help the reader to have a quick reference on the chemical structures of the reactant.

Response: We agree with the suggestion of the reviewer. The chemical formulas of

CP1, **CP1-1** and **CP1-2β** have been added in the revised Figure 1. And we have moved and modified the original Figure 1c to the revised Figure 2. (see Pages 5-6, Figs. 1-2 in the revised MS)

Comment 3: Page 6: I was not able to understand whether “mixed” crystals are formed at any of these conditions; the way the text is written, it looks like either one or the other structure of the product is formed. Three distinct reaction conditions are given, however it appears difficult to completely exclude the formation of both in the same crystal. I wonder if there is evidence that excludes the second form? Are there detailed studies of the product composition in respect to the time and temperature? As an example, see JACS (2014) 136, 558, where, due to very complex reactivity that includes various degrees of dimerization, the composition of the reacting mixture was analyzed over time.

Response: We sincerely thank the reviewer for the valuable comments. According to our previous work (*Nat. Commun.* **13**, 2847 (2022)), the conversion from **CP1** to **CP1-1** and **CP1-2β** by the combined effect of temperature and irradiation led to the formation of monocyclobutane **1** and dicyclobutane **2β**, respectively. This process can be monitored by a combination of SCXRD and ¹⁹F NMR spectra. Pure **CP1-1** can just be formed under the condition of UV light ($\lambda = 365$ nm) at -50 °C, evidenced by two peaks of chemical shift of the ¹⁹F NMR spectra observed in Supplementary Figure 3, corresponding to the signal of F-1,3-bpeb and **1**, respectively. Similarly, for the second step of the conversion from **CP1-1** to **CP1-2β** under UV light ($\lambda = 365$ nm) at room temperature, which led to two peaks of chemical shift in Supplementary Figure 4, corresponding to the signals of ligands **1** and **2β**, respectively. In contrast, if **CP1** is directly irradiated by UV light at room temperature, both **CP1-1** and **CP1-2β** could be formed, which was confirmed by the emerging of three peaks as shown in Supplementary Fig. 2. Therefore, we have revised and described the two-step [2+2] photocycloaddition reaction of **CP1** as follows:

“Although the arrangement of the C=C bonds in **CP1** should yield a monocyclobutane product upon UV light ($\lambda = 365$ nm) irradiation, the dicyclobutane

product could be obtained after **CP1** was irradiated by UV light for 1 h at 25 °C. The SCXRD analysis revealed that the basic 1D zigzag chain structure in the crystal was almost the same as that of **CP1**, and was composed of dinuclear Cd units with a dicyclobutane (**2β**), giving $[\text{Cd}_4(\mathbf{2}\beta)_2(3,5\text{-DBB})_8]$ (**CP1-2β**). Formation of **CP1-2β** suggests that one of the C=C groups in the crisscross pair in **CP1** underwent a pedal motion to give a parallel C=C pair under UV irradiation.^{43,44} Time-dependent ¹⁹F NMR spectra showed that monocyclobutane products were first formed and gradually converted to the dicyclobutane product when **CP1** was irradiated under UV light at 25 °C (Supplementary Fig. 2). When **CP1** was exposed to UV light at -50 °C, which greatly blocked the molecular rotation, only the C=C groups arranged in a parallel manner underwent dimerization while those aligned in a crisscross manner remained intact, leading to the formation of **CP1-1**. The SCXRD analysis revealed that the basic 1D zigzag chain structure in **CP1-1** was almost the same as that of **CP1**, too. **CP1-1** can be viewed as an intermediate during the formation of **CP1-2β** from **CP1**. SCXRD results, supported by ¹⁹F NMR data, indicated that UV light irradiation over **CP1** using a 2 W UV lamp ($\lambda = 365$ nm) yielded **CP1-1** in 10 min at -50 °C, which was converted to **CP1-2β** in a further 35 min irradiation at 25 °C (Fig. 2 and Supplementary Figs. 3, 4).” (see Page 7, lines 1-19 in the revised MS)

Comment 4: Page 6, the first paragraph of the results and discussion section: I am aware that that authors relate this work to a previous work and therefore the structures, except for the two molecules in Figure 1c, are not shown. However, for the readers who are not familiar with the previous work, it might be difficult to visualize and relate the discussion in this section without reference to a figure with the crystal structure that would show these details. I would suggest that the authors include panels with the relevant crystal structure fragments in Figure 1 and refer to this figure when they describe the criteria for the reactivity.

Response: We sincerely thank the reviewer for the valuable comment. According to the reviewer’s suggestion, we have added and revised the section of the structure description of **CP1** and its two-step photocycloaddition reaction as follows:

“Crystal structure and two steps of [2+2] photocycloaddition reaction. Colorless crystals of **CP1** were acquired from solvothermal reactions of $3\text{CdSO}_4 \cdot 8\text{H}_2\text{O}$ with F-1,3-bpeb and 3,5-HDBB according to the previous work.³⁸ SCXRD analysis revealed that **CP1** crystallizes in the $P\bar{1}$ space group and the asymmetric unit contains a $[\text{Cd}_2(\text{F}-1,3\text{-bpeb})_2(3,5\text{-DBB})_2]$ unit. **CP1** was previously shown to contain the diene ligands, F-1,3-bpeb, linked by Cd^{2+} ions and second carboxylate ligand 3,5-DBB to give a one dimensional (1D) zigzag chain structure. The respective configurations of the $\text{C}=\text{C}$ bonds in two adjacent F-1,3-bpeb ligands are different, resulting in one parallel and one crisscross arrangements of $\text{C}=\text{C}$ bond pairs in the chain. The distances between the parallel $\text{C}=\text{C}$ groups is 3.82 Å, which satisfies the condition for a photocycloaddition reaction to occur.⁴² The separation between the crossed $\text{C}=\text{C}$ pairs in **CP1** is 3.69 Å, but a photocycloaddition reaction requires one of the $\text{C}=\text{C}$ groups to rotate to the parallel position (Fig. 2 and Supplementary Fig. 1).

Fig. 2 | The 1D chain motifs and structures of the two-step [2+2] photocycloaddition transformation of **CP1** to **CP1-2β**. For clarity, hydrogen atoms have been omitted. Color codes: Cd, light yellow; N, blue; F, teal; Br, bright green; O, red; C, gray. The configurations and transformations of $\text{C}=\text{C}$ groups associated with the above reactions are highlighted in sky blue.

Although the arrangement of the C=C bonds in **CP1** should give a monocyclobutane product upon UV light ($\lambda = 365$ nm) irradiation, the dicyclobutane product was obtained after **CP1** was irradiated by UV light for 1 h at 25 °C. The SCXRD analysis revealed that the basic 1D zigzag chain structure in the crystal is almost the same as that of **CP1**, and is composed of dinuclear Cd units with a dicyclobutane (**2 β**), giving $[\text{Cd}_4(\mathbf{2\beta})_2(3,5\text{-DBB})_8]$ (**CP1-2 β**). Formation of **CP1-2 β** suggests that one of the C=C groups in the crisscross pair in **CP1** underwent a pedal motion to give a parallel C=C pair under UV irradiation.^{43,44} Time-dependent ¹⁹F NMR spectra showed that the monocyclobutane product was first formed and gradually converted to dicyclobutane species when **CP1** got irradiated under UV light at 25 °C (Supplementary Fig. 2). When **CP1** was exposed to UV light at -50 °C, which greatly blocked the molecular rotation, only the C=C groups arranged in a parallel manner experienced dimerization while those arranged in a crisscross manner remained intact, leading to the formation of **CP1-1**. The SCXRD analysis revealed that the basic 1D zigzag chain structure in **CP1-1** is almost the same as that of **CP1**, too. **CP1-1** can be viewed as an intermediate during the formation of **CP1-2 β** from **CP1**. SCXRD results, supported by ¹⁹F NMR data, indicated that UV light irradiation over **CP1** using a 2 W UV lamp ($\lambda = 365$ nm) yielded **CP1-1** in 10 min at -50 °C, which was converted to **CP1-2 β** in a further 35 min irradiation at 25 °C (Fig. 2 and Supplementary Figs. 3, 4).” (see Page 6, lines 2-19; Page 7, lines 1-19 in the revised MS)

Comment 5: The discussion on page 7 jumps into the changes of fluorescence in Figure 2, however, we don't see fluorescence spectra of the pure reactants/products. It would be easier to guide the reader through the expected changes in the fluorescence emission if first the spectra of the pure components were shown in Figure 2, and this would also facilitate the discussion on page 7 where the difference between the spectra are discussed.

Response: We highly appreciate the valuable suggestion of the reviewer. The solid state fluorescence excitation and emission spectra of F-1,3-bpeb, **CP1**, **CP1-1** and **CP1-2 β** have been shown in Supplementary Figure 7. According to the reviewer's suggestion,

we have added the maximum emission wavelengths (λ_{em}) and quantum yields (QYs) of **CP1**, **CP1-1** and **CP1-2 β** to guide the reader to understand the expected changes in the fluorescence emission. And these fluorescence spectra of three different components in Figs. 3b-3c have been mentioned. We have modified the manuscript as follows:

“Compounds **CP1**, **CP1-1** and **CP1-2 β** are stable in air and retain their crystalline structures intact even when immersed in ethanol for 24 h (Supplementary Fig. 6). These three compounds emitted blue light (λ_{em} 451 nm for **CP1**; λ_{em} 437 nm for **CP1-1**; λ_{em} 437 nm for **CP1-2 β**) in the solid state, with QYs of 7.8%, 58.5% and 1.2% under excitation at 365 nm at room temperature, respectively (Supplementary Fig. 7 and Supplementary Table 1).” (see Page 7, bottom to lines 2-7 in the revised MS)

Comment 6: Figure 2: please include the expressions as well as the model used to calculate the reaction order. There are different models that can be used to model solid-state reaction kinetics that can be used to obtain the order of the reaction.

Response: We are grateful to the reviewer for the valuable suggestion. In this work, the fitting of the conversion data calculated from fluorescence intensity and NMR versus irradiation time showed different kinetics for **CP1** to **CP1-1** and **CP1-1** to **CP1-2 β** , respectively. The kinetics of the transformation from **CP1** to **CP1-1** was calculated by general equation for zero-order reaction rate, while that from **CP1-1** to **CP1-2 β** was fitted by applying the Johnson-Mehl-Avrami-Kolmogorov (JMAK) model. According to the reviewer’s comment, we have added and modified the expressions of solid-state reaction kinetics calculation in the revised MS and SI as follows:

“During the transformation from **CP1** to **CP1-1**, a fitting of the conversion percentage of **CP1-1** versus UV light irradiation time at -50 °C resulted in a linear relationship of $c_0 - c_t$ with the irradiation time, indicating a zero-order behavior^{47,48} for the first step from **CP1** to **CP1-1** with a rate constant of 0.106 min⁻¹, where c_0 and c_t represent the conversion (mole fraction) calculated from fluorescence intensity data sets of **CP1** before and at any irradiation time (t) at 365 nm and -50 °C, respectively (Fig. 3e). The kinetics of the transformation from **CP1-1** to **CP1-2 β** was fitted by applying

the Johnson-Mehl-Avrami-Kolmogorov (JMAK) model, which has been successfully applied previously to a number of [2+2] photocycloadditions.^{16,24,49} The JMAK kinetics are described by Eq. 1:

$$y = 1 - e^{-(kt)^n} \quad (1)$$

where y is the conversion (mole fraction) of the photoproduct formed in time t , k is the rate constant, and n is the dimensionality of growth (Avrami exponent). The plot of $\ln(-\ln(1-y))$ versus $\ln(\text{time})$ was fitted to attain an Avrami exponent of (1.02 ± 0.03) , indicating a first-order behavior for the second step from **CP1-1** to **CP1-2 β** with a rate constant of 0.172 min^{-1} (Fig. 3f). The exponential trend in the mole ratio determined by ^{19}F NMR closely resembles those determined by in situ fluorescence intensity (Supplementary Figs. 10 - 12). In addition, a linear relationship between the conversion calculated from ^{19}F NMR and the fluorescence data for both steps could be fitted, indicating that the conversions obtained from in situ fluorescence intensity can be used similarly to those determined by ^{19}F NMR (Supplementary Fig. 13). The deviation of linearity might be due to the inhomogeneity of [2+2] photocycloaddition reaction that originated from non-uniform irradiation geometry, which was observed in the LSCM data.” (see Page 9, lines 2-25; Page 10, bottom to lines 3-14 in the revised MS)

“Kinetic analysis of each step

In order to determine the kinetics of this reaction, we monitored the corresponding structural transformation upon UV irradiation by in situ time-dependent fluorescence spectra and ^{19}F NMR. The fitting of the conversion data calculated from fluorescence intensity versus irradiation time showed different kinetics for **CP1** to **CP1-1** and **CP1-1** to **CP1-2 β** , respectively.

The kinetics of the transformation from **CP1-1** to **CP1-2 β** was fitted by applying the Johnson-Mehl-Avrami-Kolmogorov (JMAK) model.^{S3-S6} The JMAK kinetics is described by Eq. 1:

$$y = 1 - e^{-(kt)^n} \quad (1)$$

where y is the conversion (mole fraction) of the photoproduct formed in time t , k is the rate constant, and n is the dimensionality of growth (Avrami exponent).

The kinetics of the transformation from **CP1** to **CP1-1** was calculated by general

equation^{S7,S8} for zero-order reaction rate, which has been successfully applied previously to [2+2] photocycloaddition reactions. The kinetics is described by Eq. 2:

$$c_0 - c_t = kt \quad (2)$$

where c_0 and c_t represent the conversion (mole fraction) of **CP1** before and at any irradiation time (t) at 365 nm and -50 °C, respectively. k is the rate constant.” (see Page S5, bottom to lines 1-9; Page S6, lines 1-8 in the revised SI)

Comment 7: Figure 2: The plot in panel f shows significant deviation from linearity. Is there an explanation for that? Please comment in the main text.

Response: We are grateful to the reviewer for the valuable suggestion. There are many factors that may influence the photocycloaddition kinetics study, such as the heat produced by UV irradiation, the light flux of the UV lamp used, the thickness of the sample and the size of the sample particles. In Fig. 2f, the main reason for the deviation of linearity might be due to the inhomogeneity of [2+2] photocycloaddition reaction that originated from non-uniform irradiation geometry, which was observed in the LSCM data. (see Page 9, bottom to lines 1-4 in the revised MS)

Comment 8: Page 8: the sentence “while a first-order behavior fitted well for the second step” appears to be grammatically incorrect, and probably should be checked.

Response: We sincerely thank the reviewer for careful reading. The sentence “...while a first-order behavior fitted well for the second step...” has been replaced by “...indicating a first-order behavior for the second step from **CP1-1** to **CP1-2 β** with a rate constant of 0.172 min⁻¹ (Fig. 3f).” (see Page 9, lines 16-17 in the revised MS)

Comment 9: Figure 3: This may be a marginal issue, but I am not sure if it is possible to mark the ROI (regions of interest) in Figure 3c (perhaps as an inset for better visualization)?

Response: We thank the reviewer for the careful reviewing. Per the reviewer’s suggestion, we have revised and deepened the mark of ROI (regions of interest) in the revised Fig. 4c. (see Page 12, Fig. 4c in the revised MS)

Comment 10: The molecules studied in this work are very similar to the benzene dicarboxylic acid described in JACS (2009) 131, 7247, where either one or both side chains of benzene dicarboxylic acids can be dimerized, depending on the polymorph or the solvate form. I would suggest that the authors check this reference and perhaps relate their conclusions regarding the mechanisms, which has been studied using different approaches but with a similar goal in the two studies.

Response: We thank the reviewer for the valuable suggestion. Following the reviewer's suggestion, we have cited this work as Ref. 15 which described the formation of monocyclobutane and bicyclobutane by controlling the polymorph and solvate form in the revised MS. (see Page 18, Ref. 15)

Comment 11: I was intrigued by the result presented at the end of page 9 and beginning of page 10. This is indeed the central and probably the most exciting result in this work. However, the discussion here is very short and an in-depth explanation is missing. Can the authors provide deeper discussion that would relate the two clusters of data that can be seen in Figure 3e with the emission spectra of the products? Why there are only two clusters and how does that relate to the relative emission of the two products? What does that tell us of the possible different in the vertical composition of the product inside the crystal?

Response: We are grateful to the reviewer for the valuable suggestion. Figure 4e shows the fluorescence intensity change of **CP1** versus UV light ($\lambda = 365$ nm) irradiation time at 25 °C at different depths of the crystal. The fluorescence intensity in the same layer first got increased and then decreased with irradiation time, consistent with the transformations of **CP1** generating **CP1-1** and **CP1-2 β** sequentially, as shown from fluorescence spectra. Besides, the photocycloaddition reaction occurred slower in the lower layer than in the top part of the crystal. We have added a schematic of the fluorescence intensity changes in the crystal during the photocycloaddition process, to better illustrate the difference of the reaction rate at different depths. We have modified the description and Fig. 4 as detailed below:

“To gain further insight into the fluorescence intensity changes at different depths

of the crystal, we quantified the fluorescence intensities at different depths as a function of exposure time (t) by employing the z-stacked scan of LSCM which recorded a series of fluorescence snapshot images (Fig. 4e and Supplementary Fig. 16). As evident in Figs. 4e and 4f, from the 1st to the 260th slices, *i.e.*, for the top part of the crystal, the fluorescence intensity reached a maximum after 1 min of irradiation, while after the 260th slice (*ca.* 26 μm deep in the crystal), the highest fluorescence intensity was reached after 3 min of illumination, showing that the photocycloaddition reaction occurred slower in the lower layer than in the top part of the crystal. The photocycloaddition reaction in the single crystal of **CP1** first generated **CP1-1** on the top layers under UV irradiation at 25 $^{\circ}\text{C}$, accompanied by increase of fluorescence intensity in this part. As the irradiation went on, the lower part of the crystal began to gradually form **CP1-1**, and the top part was converted from **CP1-1** to **CP1-2 β** , accompanied by a brightening in fluorescence of the lower part and a darkening of the top part (Fig. 4f). Monitoring of the [2+2] photocycloaddition reaction by following the changes in fluorescence intensity clearly indicated that the [2+2] photocycloaddition reaction started from the top layers of the crystal (UV-exposed side) and gradually reached the bottom layers.

Fig. 4 | Three-dimensional (3D) LSCM tomograph of CP1. (a) Bright-field image and (b) 3D reconstitution of CP1, scale bars are 50 μm . (c) LSCM images of CP1 and (d) CP1 irradiated at 25 $^{\circ}\text{C}$ with UV light for some time intervals, at the top first slice, scale bars are 50 μm . (e) Quantified luminescence intensities in (c) ROI (regions of

interest) range for **CP1** after different irradiation times at different slices. Z represent the different slices in a single crystal. (f) Schematic illustration of the changes of fluorescence intensity in a single crystal of **CP1** under UV light irradiation at 25 °C.” (see Page 11, lines 16-30; Page 12, lines 1-10 in the revised MS)

Comment 12: The phrase “are subjected to almost the same intermolecular interactions” on page 10 sounds odd and should be rewritten. The compounds can not be subject to interactions; they do include or are composed of entities that interact with each other with intermolecular interactions.

Response: We are grateful to the reviewer for the critical comments and helpful suggestion. Per the reviewer’s suggestion, we have modified and replaced the sentence “Hirshfeld surface analysis of the minimum repetitive unit frames in **CP1**, **CP1-1** and **CP1-2β** was conducted and showed similar intermolecular interactions, indicating that these three compounds are subjected to almost the same intermolecular interactions (Supplementary Figs. 20 and 21).” by the sentence “Hirshfeld surface analysis of the structures of the repetitive units, $[\text{Zn}_2(\text{F-1.3-bpeb})_2(3,5\text{-DBB})_4]$ in **CP1**, $[\text{Zn}_2(1)(3,5\text{-DBB})_4]$ in **CP1-1** and $[\text{Zn}_2(2\beta)(3,5\text{-DBB})_4]$ in **CP1-2β**, was conducted and showed similar proportions of intermolecular interactions (Supplementary Figs. 18 and 19).” (see Page 12, bottom to lines 1-3; Page 13, line 1 in the revised MS)

Comment 13: The sentence “Although solid-state photocycloaddition reactions are well developed, the study of their dynamics and reaction mechanisms still represent a challenging field of research” is not correct. There is copious amount of literature that describes both the mechanisms and the dynamics of these reactions.

Response: We thank the reviewer for the comment and constructive suggestion. Per the reviewer’s suggestion, the sentence “Although solid-state photocycloaddition reactions are well developed, the study of their dynamics and reaction mechanisms still represent a challenging field of research” has been replaced by “Although solid-state photocycloaddition reactions are well developed, the limited availability of appropriate analytical techniques, the influence of mixtures of unreacted precursors and

intermediates, etc. make it difficult in situ monitor the progress of solid-state reactions.^{16,17} (see Page 3, lines 11-14 in the revised MS)

Comment 14: Minor point: throughout the manuscript, the word “in situ” should be written without the dash.

Response: We thank the reviewer for the comment. Per the reviewer’s suggestion, the word “*in-situ*” has been replaced by the word “in situ” in the revised MS and SI.

Reviewers' Comments:

Reviewer #2:

Remarks to the Author:

Authors have addressed review comments satisfactorily, therefore, I recommend publication of this manuscript.

Reviewer #3:

Remarks to the Author:

All of my comments have been addressed, and I now recommend the manuscript for publication in its present form.